# Differential Methylation of Telomere-Related Genes Is Associated with Kidney Disease in Individuals with Type 1 Diabetes

**DOI:** 10.3390/genes14051029

**Published:** 2023-04-30

**Authors:** Claire Hill, Seamus Duffy, Laura M. Kettyle, Liane McGlynn, Niina Sandholm, Rany M. Salem, Alex Thompson, Elizabeth J. Swan, Jill Kilner, Peter Rossing, Paul G. Shiels, Maria Lajer, Per-Henrik Groop, Alexander Peter Maxwell, Amy Jayne McKnight

**Affiliations:** 1Centre for Public Health, Queen’s University of Belfast, Belfast BT12 6BA, UK; 2Centre for Cancer Research and Cell Biology, Queen’s University of Belfast, Belfast BT9 7AE, UK; 3College of Medical, Veterinary & Life Sciences, University of Glasgow, Glasgow G12 8QQ, UK; 4Folkhälsan Institute of Genetics, Folkhälsan Research Center, 00290 Helsinki, Finland; 5Division of Nephrology, Department of Medicine, Helsinki University Central Hospital, 00290 Helsinki, Finland; 6Research Program for Clinical and Molecular Metabolism, Faculty of Medicine, University of Helsinki, 00290 Helsinki, Finland; 7Herbert Wertheim School of Public Health and Human Longevity Science, University of California San Diego, La Jolla, CA 92093, USA; 8School of Medicine, The Biodiscovery Institute, University of Nottingham, Nottingham NG7 2RD, UK; 9Nordsjaellands Hospital, Hilleroed, Denmark and Health, Aarhus University, 8000 Aarhus, Denmark; 10Steno Diabetes Center, 2730 Gentofte, Denmark; 11Department of Clinical Medicine, University of Copenhagen, 1165 Copenhagen, Denmark; 12School of Molecular Biosciences, Davidson Building, University of Glasgow, Glasgow G12 8QQ, UK; 13Department of Diabetes, Central Clinical School, Monash University, Melbourne, VIC 3800, Australia; 14Regional Nephrology Unit, Belfast City Hospital, Belfast BT9 7AB, UK

**Keywords:** biological ageing, diabetic kidney disease, epigenetic, genetic, methylation, SNP, telomere

## Abstract

Diabetic kidney disease (DKD) represents a major global health problem. Accelerated ageing is a key feature of DKD and, therefore, characteristics of accelerated ageing may provide useful biomarkers or therapeutic targets. Harnessing multi-omics, features affecting telomere biology and any associated methylome dysregulation in DKD were explored. Genotype data for nuclear genome polymorphisms in telomere-related genes were extracted from genome-wide case–control association data (n = 823 DKD/903 controls; n = 247 end-stage kidney disease (ESKD)/1479 controls). Telomere length was established using quantitative polymerase chain reaction. Quantitative methylation values for 1091 CpG sites in telomere-related genes were extracted from epigenome-wide case–control association data (n = 150 DKD/100 controls). Telomere length was significantly shorter in older age groups (*p* = 7.6 × 10^−6^). Telomere length was also significantly reduced (*p* = 6.6 × 10^−5^) in DKD versus control individuals, with significance remaining after covariate adjustment (*p* = 0.028). DKD and ESKD were nominally associated with telomere-related genetic variation, with Mendelian randomisation highlighting no significant association between genetically predicted telomere length and kidney disease. A total of 496 CpG sites in 212 genes reached epigenome-wide significance (*p* ≤ 10^−8^) for DKD association, and 412 CpG sites in 193 genes for ESKD. Functional prediction revealed differentially methylated genes were enriched for Wnt signalling involvement. Harnessing previously published RNA-sequencing datasets, potential targets where epigenetic dysregulation may result in altered gene expression were revealed, useful as potential diagnostic and therapeutic targets for intervention.

## 1. Introduction

The incidence of diabetes is increasing worldwide, with cases expected to rise to 578 million by 2030 [1]. Approximately 30 to 40% of individuals with diabetes go on to develop kidney disease [2]. In 2018, diabetes was the most common primary disease in patients requiring renal replacement therapy (RRT) in the UK and over 40% of incident end-stage kidney disease (ESKD) patients in the USA had diabetes, with significant associated healthcare costs [3,4]. 

Diabetic kidney disease (DKD) is a complex, multifactorial disease with both inherited predisposition and environmental risk factors [5,6]. Kidney disease is associated with accelerated cellular senescence and ageing [7,8,9,10]. Telomeres are nucleoprotein complexes at the ends of eukaryotic chromosomes that suffer progressive loss of nucleotides as a result of the end replication problem. This is a feature of replicative senescence and is considered a marker of ageing [7,11]. It has been associated with diseases common in older populations, including cardiovascular disease [12,13,14], diabetes [15,16,17,18,19], and chronic kidney disease (CKD) or renal dysfunction [20,21,22,23,24]. Telomere shortening in type 2 diabetes (T2D) has also been related to the presence of more disease complications [25,26,27]. The association of reduced kidney function with advancing age is well established, with studies also correlating with telomere attrition [20,22,28,29,30,31,32,33,34,35,36]. Additionally, epigenetic dysregulation is a hallmark of ageing [37] and associated with reduced renal function and telomere-related genomic instability [9]. Reduced telomerase activity has also been reported in haemodialysis patients [38].

Genome-wide association studies and exome sequencing facilitated the discovery of telomere-related genes [39,40,41,42,43,44,45,46], aiding the identification of associations between telomere length and disease susceptibility [41,46,47,48,49,50,51,52]. Few studies have explored these associations in DKD, highlighting scope to harness additional diabetic populations [16,46,53,54,55,56,57,58,59]. We investigated genomic features affecting telomere biology in patients with T1D and a consistently defined clinical phenotype of DKD in four independent White European populations, evaluating telomere length in the largest population with T1D studied to date (Figure 1).

## 2. Materials and Method

A flowchart summarising the experimental design is shown in Figure 1.

### 2.1. Study Populations

All participants provided written informed consent and were recruited from one of four populations defined in Appendix A.

### 2.2. Telomere Length

Telomere length values were determined via monochrome quantitative polymerase chain reaction (qPCR) using relative ratio of telomere repeat copy number to a single copy gene 36B4 [60] in DNA derived from whole blood. Further details are included in Appendix A.

### 2.3. Genome-Wide Association Study (GWAS)

SNPs (N = 599,104) within 376 telomere-related genes (Appendix A) were downloaded from the Ensembl genome browser using Ensembl genes 74 database on the Homo sapiens (GRCh37.p13) dataset [61]. GWAS data were obtained from the HumanOmni1-quad (Illumina Inc., California, CA, USA) for the UK-ROI cohort (n = 1830 individuals, 1,134,514 directly typed SNPs; dbGaP Study Accession: phs000389.v1.p1) [62]. In silico replication was conducted for top-ranked SNPs when considering either DKD or ESKD as the phenotype; data were retrieved from the GEnetics of Nephropathy an International Effort (GENIE) meta-analysis (dbGaP Study Accession: phs000389.v1.p1) [62]. Additional details on GWAS are included in Appendix A.

### 2.4. Mendelian Randomisation

Two-sample Mendelian randomisation was performed as described in Appendix A, harnessing SNPs utilised by Park et al. [50] and Codd et al. [44]. 

### 2.5. Epigenome-Wide Association Study (EWAS)

Existing quantitative (blood-derived) DNA methylation levels were extracted for genes relevant to telomere function from the UK-ROI cohort [63,64]. Briefly, 485,577 methylation sites for 150 DKD cases were compared to 100 controls with T1D and no evidence of kidney disease; data from 1091 CpG sites in 376 telomere-related genes were extracted for this analysis. Venn diagrams were created using the ggVenndiagram package (v. 1.2.0) in R Studio (v. 1.4.1717) [65,66]. 

### 2.6. Fine Mapping of Top Methylation Sites

Following bisulphite conversion using the EZ DNA Methylation-Lightning™ (Zymo Research, Freiburg im Breisgau, Germany), fine mapping was performed for 7 of the top-ranked differential methylation sites in MAD1L1. Primer pair sequences and PCR conditions are shown in Appendix A. Thermal cycling was performed on the MJ Tetrad PTC-225 thermal cycler with the following conditions: 15 min at 95 °C; 35 cycles of 45 s at 94 °C, 45 s at 55–65 °C (temperature optimized for each primer pair) and 1 min at 72 °C; 10 min at 72 °C. Samples were subjected to exonuclease and phosphatase clean-up and sequencing reaction using BigDye Terminator v3.1 Ready Reaction Cycle Sequencing kit (Applied Biosystems, Massachusetts, MA, USA) before cycle sequencing on the ABI 3730^®^ capillary sequencer. Results were evaluated using Contig Express on Vector NTI version 11.5.1 (Life Technologies, California, CA, USA).

### 2.7. Gene Ontology Analysis

Gene ontology analysis of biological processes, with subsequent clustering of gene ontology terms, was performed using the ViSEAGO package (v. 1.6.0), in R studio (v. 1.4.1717) [66,67], according to processes described in Appendix A.

### 2.8. PheWAS Analysis

PheWeb analysis was carried out in the UK Biobank TOPMed-imputed cohort via https://pheweb.org/UKB-TOPMed/ (accessed on 25 April 2022) (v. 1.3.15) [68]. The Common Metabolic Disease portal database [69] and the GWAS catalogue [70] were also harnessed, searching within 150 kilobases either side of the SNP co-ordinates.

### 2.9. Differential Expression Analysis

Gene expression data from previous RNA-sequencing analyses of DKD and control tissues [71,72] were analysed as described in Appendix A.

### 2.10. Statistical Analysis 

Plots in Figure 2 were created using R packages ggplot2 (v. 3.3.5) and ggpubr (v 0.4.0) [73,74]. These boxplots display the median, together with the first and third quartiles (hinges), with whiskers extending from the hinges to values at most ±1.5* interquartile range. When comparing relative telomere length (RTL) between groups, Kruskal–Wallis and Wilcoxon rank sum tests were performed (using ggpubr compare_means functions) due to the non-normality of RTL data. Where relevant, *p*-values for these tests were adjusted using the Bonferroni correction method. Where corrected RTL values are shown, RTL values were corrected for the covariates age, duration of T1D, sex, body mass index (BMI) and glycated haemoglobin (HbA1c) via a series of linear regressions, harnessing residuals as the corrected RTL values. 

## 3. Results

### 3.1. Telomere Length Associated with Premature Biological Ageing in DKD

Relative telomere length (RTL) decreased with increasing age across five age groups in the UK-ROI cohort (Figure 2A) (n = 1147; under 26 yr: 2.45 ± 1.76 (median ± interquartile range), 26 to 35 yr: 2.25 ± 1.62, 36 to 45 yr: 2.01 ± 1.40, 46 to 55 yr: 1.96 ± 1.41, over 56 yr: 1.75 ± 1.37, *p* = 7.6 × 10^−6^). Individuals with DKD had significantly shorter telomeres compared to control individuals (Figure 2B) (n = 536 cases/611 controls, *p* = 6.6 × 10^−5^). This association was ameliorated when RTL was corrected for age, T1D duration, sex, BMI and HbA1c (*p* = 0.028). When stratifying by age group, a significant difference in RTL in individuals with DKD, compared to control individuals, was only observed in the 46 to 55 age group (*p* = 0.0087) (Figure 2A). In the Steno replication cohort, however, no significant difference in telomere length (assessed by qPCR) was observed in DKD compared to controls (n = 78 cases/153 controls, *p* = 0.3). RTL in 157 cases and 116 controls from this population were previously measured using Southern blot [56]. The correlation between the qPCR and Southern blot (SB) results (r > 0.8) highlighted the consistency of methods used [75,76].

### 3.2. Genetic Variants Were Nominally Associated with Telomere Length

Existing genome-wide SNP data were used to investigate association with RTL in 1019 individuals from the UK-ROI cohort, adjusting for age (Appendix A). Nominally associated hits (*p* < 0.005) are shown in Appendix A. Of the SNPs within the 376 telomere-related genes (Appendix A), no SNP reached significance (*p* = 10^−8^); however, 36 SNPs in 22 genes demonstrated nominal association (Table 1 and Appendix A). In total, 17 of these SNPs were within genes nominally associated with hypertension, diabetes, renal, glomerulonephritis or nephritis-related phenotypes in the UK Biobank (Appendix A). Moreover, searching within 150 kb of these SNP co-ordinates revealed that 25 of these regions (69.4%) were significantly associated with a range of renal or cardiovascular phenotypes in the Common Metabolic Disease portal database (clumped by linkage disequilibrium) [69], which aggregates and analyses human genetic and functional genomic information linked to common metabolic diseases from up to 398 datasets (Appendix A). Using these same regions to search the GWAS Catalogue [70], it was shown that the rs852540 SNP region (7:5,383,963–5,733,963) contained the variant rs7808152, previously associated with telomere length (*p* = 1 × 10^−6^, Beta = −0.1602507, CI = 0.096–0.225) in a cohort of 902 European-ancestry individuals (Netherlands) [77]. It was also shown that, within 150 kb of variants rs2209437, rs2025557 and rs1536078 (all with the closest gene *SH3GL2*), variants associated with DNA methylation (rs7032102, *p* = 2 × 10^−8^) and epigenetic clock age acceleration (GrimAge) (rs1114790, *p* = 10^−8^, Beta = −1.0232, CI = 0.67–1.38) were identified in a cohort of up to 954 individuals from the UK [78]. The highest-ranked telomere-related SNP was rs2725385 in the *WRN* gene (*p* = 2.09 × 10^−4^, OR = 1.28, 95% CI = 1.22–1.45). An additional 5 of the top 36 SNPs were also located in this gene, all within strong linkage disequilibrium (D´ > 0.8).

### 3.3. Genetic Variants within Telomere-Related Genes Were Nominally Associated with DKD and ESKD

Using existing GWAS data, SNPs within the telomere-related genes were investigated for an association with DKD and ESKD in the UK-ROI cohort. In total, 6582 SNPs within telomere-related genes were present within the discovery GWAS. Comparison of 823 individuals with DKD (cases) with 903 individuals with T1D but no evidence of kidney disease (controls), corrected for age, sex, and duration of diabetes, revealed no genome-wide significant SNPs; however, 28 SNPs in 14 genes demonstrated nominal association (Table 2). The highest-ranked SNP was rs2299694 in *ADA* (*p* = 9.77 × 10^−5^, OR = 1.83, 95% CI = 1.35–2.47). All SNPs reaching nominal association underwent in silico replication via meta-analysis with the FinnDiane (n = 2910) and US GoKinD (n = 1595) datasets. Association was supported for two SNPs (Table 2), the most significant being rs2292681 in *RNF10* (*p* = 2.81 × 10^−3^). 

Analysis for the ESKD phenotype in 247 cases and 1479 controls did not reveal significant SNPs; however, 45 SNPs in 17 unique genes were nominally associated. In silico replication supported nine of these SNPs (Table 3), including SNPs within the top-ranked genes, *PIPOX* (rs7220474, *p* = 0.047) and *DPP3* (rs2279863, *p* = 0.028 and rs1671063, *p* = 0.013).

### 3.4. Exploring Mendelian Randomisation to Inform Associations between Genetic Telomere Length and DKD

Existing meta-analysis of genetic variants associated with kidney disease in T1D was used as the outcome dataset, considering either DKD or ESKD. As instrumental variables (IVs), 130 variants shown by Codd et al. to be significantly associated with leukocyte telomere length were utilised [48]. Separately, 33 variants used by Park et al. to highlight an association between telomere attrition and CKD were harnessed [47,50]. When utilising either set of IVs, no significant association between telomere length and DKD was identified, with no significant pleiotropy present (Table 4 and Appendix A). Whilst no significant association was identified between ESKD and telomere length, significant pleiotropy was identified when utilising the Codd et al. IVs (Table 4 and Appendix A). The MR-PRESSO method confirmed the presence of pleiotropy (global test *p* = 0.0193) [79]; however, no outlying variants were identified and the causal effect was not significantly distorted. 

### 3.5. Significant Differential Methylation of Telomere-Related Genes in DKD and ESKD

Focusing on epigenetics, 496 methylation sites in 212 genes reached epigenome-wide significance (*p* ≤ 10^−8^) for DKD (Appendix A). cg00445824 in *ISYNA1* was the most significantly associated site (*p* = 9.1 × 10^−24^), with this site also significant and in the same direction for ESKD. ESKD was associated with 412 methylation sites in 193 unique genes at epigenome-wide significance (Appendix A). The most significant was cg19898668 in *REM2* (*p* = 2.2 × 10^−21^), with this site also significant in the same direction for DKD. Four genes (*MAD1L1, TBCD, BANP* and *PFKB*) contained significant differential methylation at more than 10 sites for DKD and ESKD (Table 5). ESKD beta value distributions of top-ranked sites in these genes are shown in Figure 3. Venn diagrams comparing the CpG sites (Appendix A) and associated genes (Appendix A) show that, whilst 40% of differentially methylated sites overlap between DKD and ESKD, 70% of differentially methylated genes overlap between these phenotypes. Of the top 15 genes presenting differential methylation in both phenotypes (Table 5), *MAD1L1, TUBB*, *HIST1H2AL* and *TBCA* were significantly associated with diabetes-related phenotypes in the UK Biobank (Table 6). *MAD1L1* was fine mapped for top-ranking methylation sites (Figure 4). 

Gene ontology analysis of the biological processes enriched in the significantly differentially methylated telomere-related genes, compared to the full telomere-related profile, revealed associations with telomere and chromosomal maintenance. Differential methylation within genes with predicted roles in Wnt signalling was also observed for DKD and ESKD (Appendix A). Morphogenesis, biosynthetic processing and metabolism regulation were additionally highlighted for DKD alone (Appendix A).

### 3.6. Gene Expression Changes in Telomere-Related Genes Reflected Differential Methylation Patterns

To explore the effects of differential methylation during DKD, methylation data were compared with gene expression data from micro-dissected glomerular and tubulointerstitial tissue [71]. The resultant log fold change (logFC) data from a differential expression analysis in DKD, compared to living donor controls, were correlated with mean delta-beta values for the significantly differentially methylated CpG sites within each gene (Figure 5). Some genes displayed a change in methylation consistent with their expression pattern during DKD (with fold change greater than 2 in the increasing or decreasing direction); Figure 6 explores these genes further. Interestingly, a number of genes which presented directional change in transcriptomic data, which was concordant with the differential methylation in DKD or ESKD, were associated with hypertension, diabetes, renal, glomerulonephritis or nephritis-related phenotypes in the UK Biobank (Table 7).

An additional RNA-sequencing dataset generated from the whole-kidney biopsies of control, early DKD and advanced DKD participants was harnessed [72]. LogFC values for genes of interest in this analysis were generally lower and, therefore, a fold change cut-off of greater than ±1 was utilised (Appendix A). *CAMK1D, ENO2, FANCA, MTX1* and *DOK2* displayed consistency between datasets (Appendix A, Figure 5 and Figure 6). For *HSPA6*, decreased expression occurred during early DKD, mirroring the first dataset; however, this study revealed a subsequent increase in *HSPA6* expression during advanced DKD, which may better reflect the decreased methylation associated with ESKD. 

When assessing the expression level of telomere-related genes with the highest number of statistically significant methylation sites (Table 5) or those genes with predicted association with Wnt signalling, all genes except WIPI2 showed a significant differential expression in at least one comparison (Appendix A). Interestingly, *C12orf43*, *TBL1X* and *TNKS* were significantly associated with diabetes or hypertension-related phenotypes in the UK Biobank (Appendix A).

## 4. Discussion

In the present study, we explored the association of telomere length with DKD and ESKD in a European T1D population, determining RTL in 1147 participants. Mean RTL was significantly shorter in individuals with DKD, compared to individuals with T1D and no evidence of kidney disease, even after covariate correction. Whilst shorter telomere length has been associated with a significantly increased risk of DKD in diabetes (T1D and T2D combined) [54], mixed associations between telomere length and renal function have been reported. A systematic review by Ameh et al. has investigated associations between telomere length and renal traits in 7,829 individuals from nine studies, two specifically investigating diabetic patients (T1D [53] and T2D [25]) with varying stages of kidney disease [20,25,53]. Telomere attrition was associated with estimated glomerular filtration rate (eGFR) decline and kidney disease progression among people with diabetes [20]. However, longer telomeres were associated with longer kidney disease duration, perhaps influenced by telomere repair in longer surviving CKD patients [20]. Fyhrquist et al. have highlighted that short telomeres were independent predictors of DKD progression in patients with T1D [53]. A recent longitudinal study by Syreeni et al. has revealed that individuals with T1D with the shortest telomeres had lower eGFR, increased albuminuria and more stage 3 CKD [59]. Januszewski et al., however, determined that, whilst telomeres were shorter in patients with T1D versus a control group, RTL did not correlate significantly with renal function [58]. A recent review from our group highlights the recent literature connecting telomere length and DKD [80]. 

Telomere length does not correlate well with all aspects of renal function, especially after age adjustment [21,22,49,55,81,82]. Sun et al. determined that SNPs in telomere-related genes, rather than telomere length, contributed to primary glomerulonephritis/ESKD susceptibility [49]. Other studies determined that genetic telomere length was not significantly associated with CKD or diabetes, perhaps due to residual biases or limited power [44,47]. These data reflect that telomere length is a weak marker of ageing, displaying substantial inter-individual variation, reflecting differing exposomes [83]. For example, variation in ageing linked to renal function is also strongly influenced by diet and the microbiome, which may confound primary analyses [84]. This highlights the need for multi-omic studies to assess additional factors, such as epigenetics, which can be altered during the life course [5,64,85]. We therefore conducted, to our knowledge, the first investigation of leukocyte telomere length, together with both genetic and epigenetic status of nuclear telomere-related genes, for association with DKD in T1D. We used cost-effective methods to determine telomere length and high-density microarrays for efficient and high-throughput examination of telomere-related gene data extracted from ~1 million SNPs and ~480,000 methylation sites at single-base resolution. 

Using existing genotype data [62], SNPs were investigated for an association with qPCR-established telomere length, DKD and ESKD. A total of 17% of the top 36 SNPs associated with telomere length were present in *WRN*. Variants in this gene are the primary cause of Werner syndrome, a disorder of accelerated ageing, with *WRN* knockout cell lines presenting with accelerated telomere attrition [86,87,88]. In vitro studies have demonstrated that the WRN helicase interacts with TERF2, a member of the shelterin complex essential for telomere maintenance [87,89]. TERF1 is an additional protein implicated in WRN function, with a variant near TERF1-interacting nuclear factor 2 (*TINF2*) associated with telomere length and T2D, and is shown to drive the association between telomere length and CKD in T2D [51,52,90]. Together, these studies highlight the potential for WRN functioning and TERF1- or TERF2-interacting proteins to influence the risk of DKD. It is important to note, however, that no SNP reached genome-wide significance when assessing RTL, and genes such as *LMNA*, implicated in telomere stability and CKD [91,92], were not identified, highlighting the need for even larger genetic datasets of this type to increase the power and uncover novel gene–function interactions.

The nominal association with DKD was supported during replication analysis for two SNPs, the most significant being rs2292681 in *RNF10*. Liu et al. recently showed via a multi-ancestry meta-analysis of 1.5 million individuals that rs3817301 in *RNF10* was significantly associated with eGFR [93]. For ESKD, variants within seven genes were supported during replication analysis, with variants within *SYK* and *PIPOX* previously associated with T2D or HbA1c [94,95]. *PIPOX* expression was shown to significantly decrease during DKD [96], encoding a protein involved in maintaining oxidative stress balance [96], a process disrupted during DKD [97] and a key factor in modulating telomere shortening [98].

Utilising Mendelian randomisation, we assessed the causal association between genetically determined telomere shortening and DKD or ESKD, harnessing two sets of IVs previously used to explore kidney disease in European populations [47,48,50]. Park et al. determined that telomere shortening was significantly associated with a higher CKD risk (OR 1.20, *p* < 0.001), with successful replication in the UK Biobank [47,50]. Li et al. utilised the same genetic instrument in a UK Biobank cohort to reveal that genetically-determined telomere attrition did not affect the risk of diseases such as diabetes or CKD [47]; however, Park et al. highlighted that this null result may be due to the low number of self-reported or ICD-confirmed diagnosis for CKD and instead opted for serum cystatin C and creatinine level classification [50]. Codd et al. identified IVs of leukocyte telomere length in a UK Biobank cohort and, whilst these authors determined that direct measurements of telomere length were significantly associated with CKD within the UK Biobank (Hazard ratio = 0.889, *p* value = 9.45 × 10^−17^), genetically determined telomere length was not (OR = 1.02, *p* = 0.71), with CKD classified based on self-reported, ICD and procedure codes [48]. Harnessing these genetic instruments for analysis within our cohort revealed no significant associations between telomere shortening and DKD or ESKD. Additional studies have explored the effect of genetically determined telomere length and kidney disease. Gurung et al. demonstrated that genetically determined telomere attrition was associated with increased DKD in East Asian T2D patients [52]. Taub et al. highlighted that rs1008438 in *HSPA1A* gene was significantly associated with DKD risk in T1D [46]. These results highlight the complexity of the association between telomere length and kidney disease, with results in both genetic-based and observational studies dependent on the measure of kidney function used [22]. It is important, therefore, to take a wider approach, assessing both the observational and genomic nature of telomere length to gain a fuller understanding of kidney disease pathogenesis.

Via a multi-omics approach, we broadened our study of telomere-related genes in DKD. Multiple statistically significant methylation sites were identified in *MAD1L1*, encoding a protein involved in the mitotic spindle assembly, chromosome alignment, cell cycle control and tumour suppression. *MAD1L1* inhibits *TERT* transcription via epigenetic modification of histones [99]. *TERT* encodes the catalytic subunit of telomerase and plays a crucial role in telomere maintenance and senescence [100]. Methylation of *TERT* and *MAD1L1* in DKD may alter telomerase activity, potentially reducing the regenerative ability of renal cells [101,102]. Via TERT, telomerase can also modulate Wnt/β-catenin signalling [103]. This pathway is important for podocyte proliferation, the epithelial cells involved in maintaining normal kidney filtration (Figure 7). Interestingly, gene ontology analysis revealed that Wnt signalling was an enriched biological process within the telomere-related genes which were significantly differentially methylated in DKD or ESKD. Altering DNA methylation, even via dietary interventions [84,104], may prove a novel therapeutic opportunity for DKD [105], with this analysis identifying targets for future study (*TERT, VAX2, C12orf43, TBL1X, TNKS, BCL7B, ZEB2* and *AKT1*). 

A limitation of this study was the analysis of telomere length and epigenetics at a single time point. A longitudinal study, as carried out by Syreeni et al. in a smaller T1D cohort [59], would provide insights into disease progression and (epi)genomic changes over time. In addition, incorporating assessment of individual exposome factors, such as lifestyle, diet, environment and socioeconomic position, as well transgenerational exposome stressors, may also be of merit, due to their impact on accelerated ageing, kidney disease and the epigenome [83]. 

Employing previously published RNA-sequencing datasets generated using human kidney tissue biopsies [71,72], differential methylation of telomere-related genes during DKD and ESKD was correlated with altered gene expression. Differentially methylated genes with predicted roles in Wnt signalling were significantly differentially expressed during DKD; however, the extent of this differential expression was limited. Differentially methylated genes showing a concordant change in gene expression during DKD were identified. Genetic variants within *BICD1* were nominally associated with DKD and ESKD in the discovery and replication GWAS, with rs7900065 within *BAG3* nominally associated with ESKD in both datasets. A *BICD1* genetic variant was previously associated with longitudinal DNA methylation changes in an older population [78]. The present study also highlighted a telomere-related gene (*SH3GL2*), which contained genetic variants nominally associated telomere length and genetic variants within 150 kilobases previously associated with DNA methylation or accelerated epigenetic ageing. Interestingly, *SH3GL2* was significantly differentially methylated in ESKD in this study and significantly dysregulated in both the Fan et al. and Levin et al. RNA-sequencing datasets, with decreased expression observed alongside (on average) decreased methylation. These results highlight the potential interaction between genetic variation in telomere-related genes, methylation changes and gene expression, which may impact kidney health during T1D, identifying potential diagnostic and therapeutic targets.

Whilst a number of genes with a high number of differentially methylated sites, such as *MAD1L1*, presented significant differential expression, the extent of this differential expression was limited. Therefore, specifically methylated sites, rather than an accumulation of differentially methylated sites, may influence gene expression during DKD. Future work is needed to explore the effects of differential methylation in DKD and ESKD, especially for genes where extensive transcriptional change was not observed or where gene expression patterns were not concordant with methylation changes.

## 5. Conclusions

In summary, taking a multi-omic approach, telomere-related genes nominally associated with telomere length, DKD, or ESKD were identified, with epigenetic analysis revealing approximately 230 genes differentially methylated in DKD and/or ESKD. These differentially methylated genes were enriched for Wnt signalling functions. Harnessing previously published transcriptomic datasets, differential methylation was correlated with gene expression in DKD, highlighting potential targets where epigenetic dysregulation may result in altered gene expression and influence disease. This study identified potential targets where epigenetic regulation of telomere function may have functional consequence on DKD, useful as potential diagnostic and therapeutic targets for intervention.

## Figures and Tables

**Figure 1 genes-14-01029-f001:**
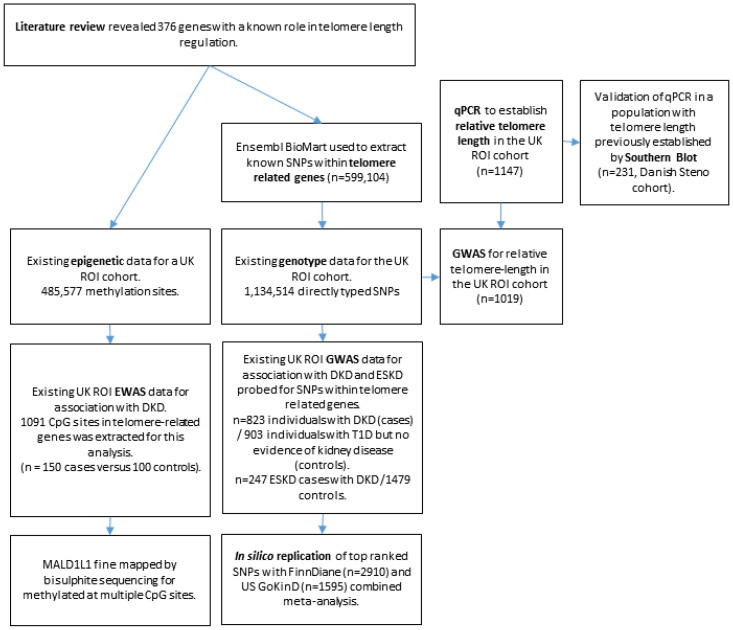
Flowchart showing experimental design of the project. DKD = diabetic kidney disease. ESKD = end-stage kidney disease.

**Figure 2 genes-14-01029-f002:**
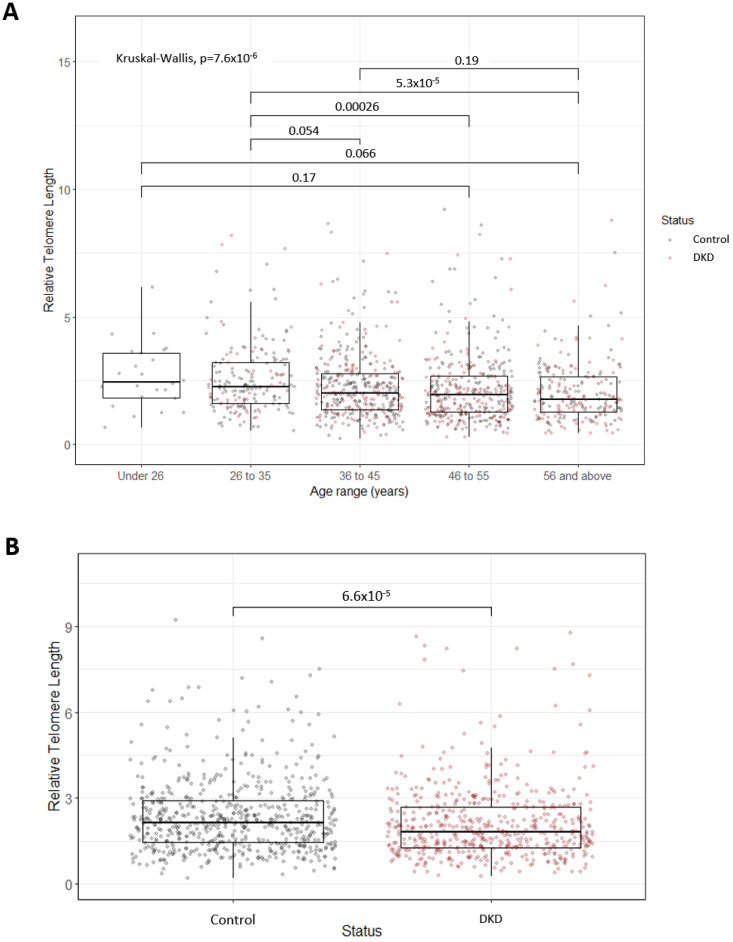
(**A**) Boxplots representing decreasing relative telomere length with advancing age. The P-value for the overall Kruskall–Wallis test highlighted significant differences between age groups. Pairwise comparisons using Wilcoxon tests were also carried out, with significant adjusted P-values displayed (Bonferonni correction). Points are coloured based on diabetic kidney disease (DKD) status. The numbers of individuals in each age group are as follows: Under 26, N = 24; 26 to 35, N = 181; 36 to 45, N = 369; 46 to 55, N = 359; 56 and above, N = 214. (**B**) Boxplots representing the significant difference in relative telomere length between individuals with DKD and control individuals with at least 15 years duration of T1D but no evidence of kidney disease. Significance was determined via a Wilcoxon test.

**Figure 3 genes-14-01029-f003:**
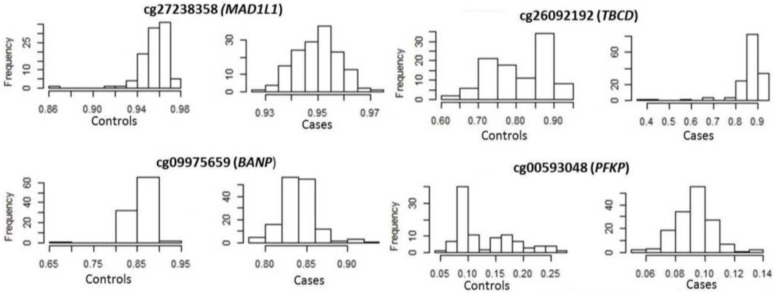
Distribution of methylation of top-ranked sites in end-stage kidney disease cases and controls.

**Figure 4 genes-14-01029-f004:**
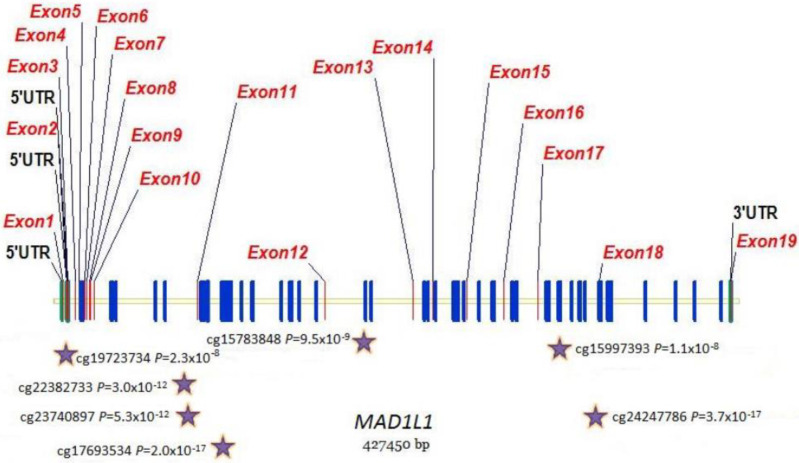
Distribution of methylation sites across the MAD1L1 gene. Stars show top-ranked sites that were fine mapped.

**Figure 5 genes-14-01029-f005:**
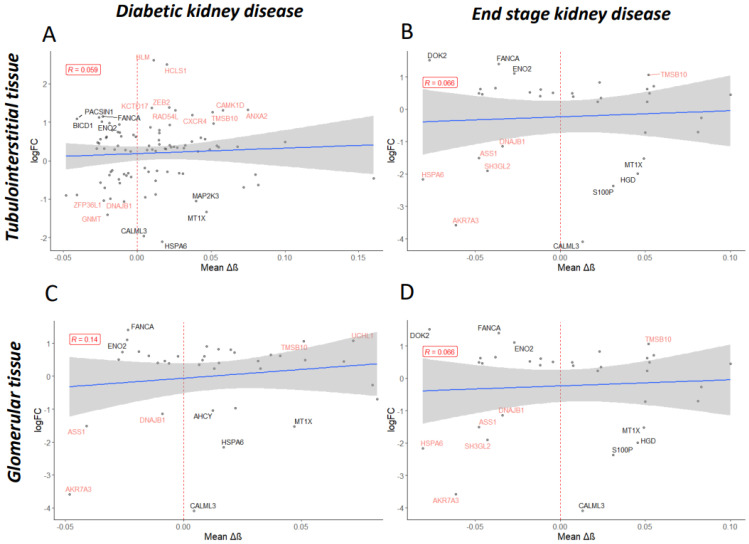
Mean delta-beta (Δβ) values per gene were determined for the differentially methylated CpG sites significantly associated with diabetic kidney disease (**A**,**C**) or end-stage kidney disease (**B**,**D**). Mean Δβ values were regressed on log fold change (logFC) values obtained during the differential expression analysis of RNA-sequencing data for tubulointerstitial (**A**,**B**) or glomerular (**C**,**D**) tissue from patients diagnosed with diabetic nephropathy versus controls (Levin et al.). Only genes shown to be significantly dysregulated (*p*-adj < 0.05) are displayed. Shown in black are genes of interest where delta-beta methylation values correlate accordingly with expression level changes (for example, increased methylation correlates with decreased expression); shown in red are genes which do not. Labelled are genes with ±1 logFC.

**Figure 6 genes-14-01029-f006:**
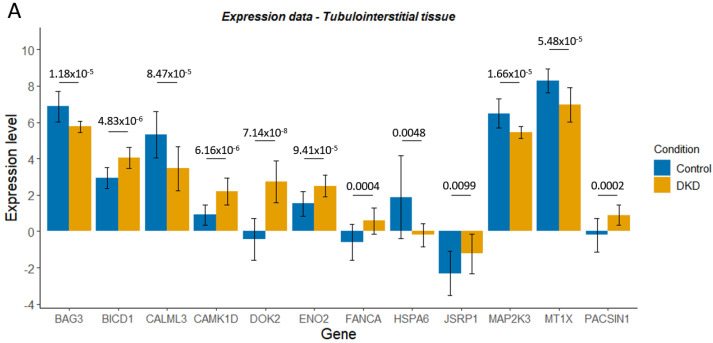
Expression level (Levin et al., normalised log2 read count) for control and diabetic kidney disease (DKD) tubulointerstitial (**A**) or glomerular (**B**) tissue for genes of interest. Delta-beta (Δβ) values for the CpG sites within these genes of interest, whose differential methylation was significantly associated with DKD or end-stage kidney disease (ESKD) (**C**). *ns* denotes genes that were not significantly differentially methylated in DKD/ESKD. Mean ± SD is shown. In graphs A and B, p-adjusted values from the differential expression analysis are shown.

**Figure 7 genes-14-01029-f007:**
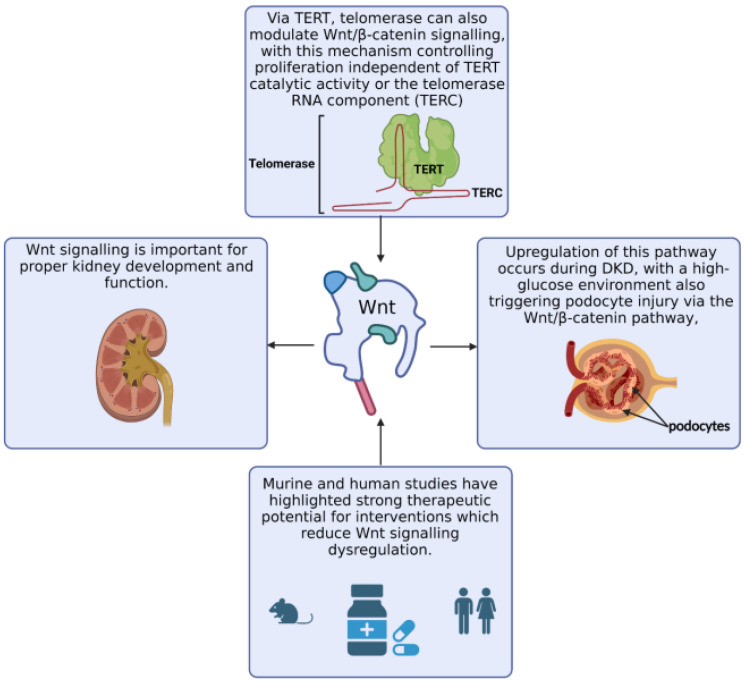
Via TERT, telomerase can also modulate Wnt/β-catenin signalling, with this mechanism controlling proliferation independent of TERT catalytic activity or the telomerase RNA component (TERC) [103,106,107]. Wnt signalling influences on kidney development and functioning, with potential for this pathway to be targeted for therapeutic interventions for DKD [108,109,110,111,112,113,114,115,116,117].

**Table 1 genes-14-01029-t001:** Genes with multiple SNPs demonstrating nominal association (*p* < 0.005) with telomere length.

Gene	Number of SNPs with *p* < 0.005	Top Ranked SNP	Top Ranked SNP *p*-Value	Top Ranked SNP MAF	Top Ranked SNP Odds Ratio
*WRN*	6	rs2725385	2.09 × 10^−4^	0.29	1.28
*HAAO*	3	rs11891403	8.17 × 10^−4^	0.48	1.26
*NUDCD2*	3	rs6877450	1.94 × 10^−3^	0.05	1.49
*SH3GL2*	3	rs2209437	1.97 × 10^−3^	0.43	0.83
*PFKP*	2	rs10795000	6.59 × 10^−4^	0.11	1.39
*RAD51B*	2	rs17105494	1.13 × 10^−3^	0.12	1.35
*PAK4*	2	rs12976130	2.60 × 10^−3^	0.39	0.83

**Table 2 genes-14-01029-t002:** SNPs in telomere-related genes nominally associated with diabetic kidney disease (DKD) in the discovery cohort, with their association with DKD in the replication cohort also shown. Highlighted in green are those genes significantly associated with DKD in the replication cohort. CHR = chromosome, BP = base pair, A1 = allele 1, OR = odds ratio, SE = standard error, L/U95 = lower/upper 95% confidence intervals.

					Discovery					Replication			
SNP	Gene	CHR	BP	A1	OR	SE	L95	U95	*p*-Value	Harmonised OR	SE	*p*-Value	Direction
rs2299694	*ADA*	20	42696612	G	1.825	0.1544	1.348	2.47	9.77 × 10^−5^	1.1391	0.0729	0.07403	--+
rs9938618	*MT3*	16	55184270	G	2.17	0.2066	1.447	3.253	0.000178	1.2215	0.0959	0.03699	--+
rs10844194	*BICD1*	12	32424997	A	1.614	0.1373	1.233	2.112	0.000493	1.0887	0.0644	0.1864	++-
rs11051966	*BICD1*	12	32421847	A	1.612	0.1372	1.232	2.109	0.000504	1.0904	0.0628	0.1685	++-
rs11655449	*CRK*	17	1274465	A	0.689	0.1092	0.5563	0.8535	0.000648	0.9459	0.0433	0.1994	+-+
rs10771942	*BICD1*	12	32419755	A	1.357	0.09409	1.129	1.632	0.001173	1.0285	0.0392	0.4735	-+-
rs1060419	*HMGN3*	6	80001713	A	0.7325	0.09609	0.6067	0.8843	0.001197	0.9663	0.039	0.3796	--+
rs7298381	*BICD1*	12	32421524	T	1.346	0.0948	1.117	1.62	0.001738	1.0272	0.0394	0.4958	++-
rs2292681	*RNF10*	12	1.19E+08	G	0.7265	0.1022	0.5945	0.8876	0.001773	0.8854	0.0407	0.002811	+++
rs2306042	*SYK*	9	92680972	C	3.658	0.4154	1.621	8.257	0.001793	NA	NA	NA	NA
rs2920881	*RTN4*	2	55152021	T	1.423	0.1132	1.14	1.777	0.001821	NA	NA	NA	NA
rs7964583	*BICD1*	12	32414859	A	1.353	0.09736	1.118	1.638	0.001897	1.0758	0.0401	0.06839	+++
rs1324646	*PROSER2*	10	11917712	A	1.469	0.1258	1.148	1.88	0.002241	1.0933	0.0539	0.09788	+++
rs2304661	*HAAO*	2	42847988	T	0.5062	0.2242	0.3262	0.7855	0.002389	0.9096	0.0891	0.2882	+-+
rs9352704	*HMGN3*	6	79967037	C	0.7372	0.1017	0.604	0.8998	0.00271	0.9499	0.0403	0.202	++-
rs2415251	*RPP25*	15	73029208	T	1.335	0.0969	1.104	1.614	0.002899	1.0779	0.0394	0.05675	+++
rs7076760	*PROSER2*	10	11921260	G	1.448	0.1251	1.133	1.85	0.003096	1.0910	0.0534	0.1031	+--
rs12214161	*HMGN3*	6	80004995	G	1.324	0.09517	1.099	1.596	0.003172	1.0463	0.0394	0.2498	--+
rs11551942	*RTN3*	11	63283277	A	1.464	0.1293	1.136	1.886	0.003225	1.0176	0.0596	0.7707	-+-
rs11655637	*CRK*	17	1268298	T	0.7507	0.09753	0.6201	0.9088	0.003275	0.9424	0.0402	0.14	+--
rs13428739	*EPAS1*	2	46377438	T	1.705	0.1814	1.195	2.433	0.003283	NA	NA	NA	NA
rs155202	*BICD1*	12	32398124	G	1.314	0.09427	1.092	1.581	0.003773	0.9847	0.0393	0.6957	+-+
rs7959002	*BICD1*	12	32395778	T	1.314	0.09425	1.092	1.58	0.003787	0.9835	0.0393	0.6727	-+-
rs2540477	*LTA4H*	12	94961685	G	0.7128	0.1177	0.5659	0.8977	0.004017	NA	NA	NA	NA
rs6416877	*CRK*	17	1313872	T	0.7273	0.1107	0.5855	0.9036	0.004028	0.9383	0.0436	0.1435	---
rs7314141	*BICD1*	12	32421314	C	1.323	0.09867	1.09	1.605	0.004569	1.0761	0.0404	0.06963	--+
rs7208768	*CRK*	17	1299451	G	0.7613	0.09685	0.6297	0.9204	0.004862	0.9338	0.0401	0.08744	+++
rs11657524	*CRK*	17	1304873	C	0.7613	0.09701	0.6295	0.9207	0.004936	0.9270	0.04	0.05818	---

**Table 3 genes-14-01029-t003:** SNPs in telomere-related genes nominally associated with end-stage kidney disease (ESKD) in the discovery cohort, with their association with ESKD in the replication cohort also shown. Highlighted in green are those genes significantly associated with ESKD in the replication cohort. CHR = chromosome, BP = base pair, A1 = allele 1, OR = odds ratio, SE = standard error, L/U95 = lower/upper 95% confidence intervals.

					Discovery					Replication			
SNP	Gene	CHR	BP	A1	OR	SE	L95	U95	*p*-Value	Harmonised OR	SE	*p*-Value	Direction
rs7207327	*PIPOX*	17	24375852	T	1.871	0.1546	1.382	2.533	5.06 × 10^−5^	1.080	0.0581	0.1844	-++
rs7220474	*PIPOX*	17	24365560	A	1.983	0.1693	1.423	2.763	5.25 × 10^−5^	1.141	0.0663	0.0474	-++
rs2511224	*DPP3*	11	66019182	T	0.5419	0.1553	0.3997	0.7348	8.01 × 10^−5^	0.912	0.0537	0.08572	---
rs2279863	*DPP3*	11	66004272	T	1.74	0.1454	1.309	2.314	0.000139	1.122	0.0524	0.02767	+++
rs1671063	*DPP3*	11	66028718	A	1.726	0.1451	1.299	2.294	0.000168	1.138	0.0524	0.01333	+++
rs17759	*PHYKPL*	5	178000000	A	1.779	0.1557	1.311	2.414	0.000216	1.087	0.0608	0.1682	++-
rs2015918	*PIPOX*	17	24375223	A	1.822	0.1656	1.317	2.521	0.00029	1.109	0.069	0.1322	++-
rs997996	*PIPOX*	17	24375446	G	1.818	0.1658	1.313	2.515	0.000312	1.109	0.069	0.1348	--+
rs6830321	*ANXA5*	4	123000000	T	0.604	0.1418	0.4574	0.7975	0.000376	0.946	0.0502	0.2669	+-+
rs17577590	*PHYKPL*	5	178000000	A	1.706	0.1509	1.269	2.294	0.000399	1.117	0.0563	0.04865	++-
rs1324646	*PROSER2*	10	11917712	A	1.868	0.1798	1.313	2.657	0.000511	1.063	0.0701	0.3859	-++
rs4795487	*PIPOX*	17	24368556	A	1.725	0.1606	1.259	2.363	0.000686	1.098	0.0624	0.1335	-++
rs3136814	*APEX1*	14	19993137	C	2.691	0.3001	1.494	4.846	0.000974	1.469	0.1603	0.01644	---
rs11831773	*BICD1*	12	32202114	T	0.394	0.2832	0.2262	0.6864	0.001006	0.907	0.0811	0.2271	---
rs7076760	*PROSER2*	10	11921260	G	1.8	0.1787	1.268	2.555	0.001012	1.054	0.0693	0.4443	+--
rs10937831	*LYAR*	4	4323403	T	1.594	0.1426	1.206	2.109	0.001069	0.995	0.0545	0.9249	-+-
rs1844754	*PIPOX*	17	24365902	G	1.827	0.1872	1.266	2.637	0.001283	1.305	0.0816	0.001113	---
rs11098637	*ANXA5*	4	123000000	T	0.6328	0.1435	0.4777	0.8383	0.001426	0.936	0.0504	0.1899	---
rs2292542	*AIPL1*	17	6278948	T	4.906	0.5017	1.835	13.12	0.001524	2.006	0.2807	0.01316	++?
rs3749813	*PHYKPL*	5	178000000	A	1.725	0.1754	1.223	2.433	0.001879	1.069	0.0663	0.3114	-+-
rs7964583	*BICD1*	12	32414859	A	1.555	0.1426	1.176	2.057	0.001956	1.092	0.0518	0.08982	+++
rs2306420	*ANXA5*	4	123000000	A	1.58	0.1482	1.182	2.112	0.002027	1.094	0.0552	0.1023	++-
rs2299743	*PCP4*	21	40162449	C	0.5789	0.1778	0.4086	0.8202	0.002107	0.920	0.0593	0.1598	++-
rs643788	*H2AFX*	11	118000000	C	0.6404	0.145	0.4819	0.8509	0.002115	0.974	0.0511	0.6065	-++
rs2509049	*H2AFX*	11	118000000	T	0.6407	0.1449	0.4823	0.8512	0.002129	0.976	0.0511	0.6312	+--
rs8551	*H2AFX*	11	118000000	T	0.6416	0.1449	0.4829	0.8524	0.002201	0.975	0.0511	0.6256	+--
rs3771395	*VAX2*	2	70986522	A	1.776	0.1891	1.226	2.573	0.002382	0.991	0.094	0.924	-+-
rs2282640	*DPP3*	11	66002427	T	0.6161	0.1599	0.4503	0.8429	0.002455	0.903	0.0573	0.07525	---
rs2306042	*SYK*	9	92680972	C	4.929	0.5268	1.755	13.84	0.002459	1.808	0.2384	0.01299	---
rs1058640	*BICD1*	12	32406991	A	1.531	0.1415	1.16	2.021	0.002614	1.062	0.051	0.24	-++
rs13145977	*ANXA5*	4	123000000	C	1.557	0.1475	1.166	2.079	0.002675	1.098	0.0554	0.08973	--+
rs155202	*BICD1*	12	32398124	G	1.519	0.1405	1.154	2.001	0.002911	1.000	0.0507	0.9957	+-+
rs7959002	*BICD1*	12	32395778	T	1.519	0.1405	1.153	2	0.002919	0.998	0.0507	0.9692	-+-
rs7208041	*YWHAE*	17	1217312	G	1.651	0.17	1.183	2.303	0.003197	1.000	0.0646	0.9977	--+
rs8078073	*YWHAE*	17	1215433	C	1.651	0.17	1.183	2.303	0.003197	1.010	0.0647	0.8825	--+
rs7900065	*BAG3*	10	121000000	G	2.305	0.2838	1.322	4.021	0.003253	1.242	0.1228	0.07702	--+
rs1865328	*LYAR*	4	4318830	G	0.6517	0.1459	0.4897	0.8675	0.003341	0.987	0.0579	0.819	-++
rs11655979	*PIPOX*	17	24388404	T	0.5895	0.1817	0.4129	0.8417	0.003636	0.954	0.058	0.4163	--+
rs16945809	*YWHAE*	17	1241236	C	2.029	0.2443	1.257	3.276	0.003773	1.107	0.1199	0.3952	--+
rs325429	*BICD1*	12	32405382	G	1.496	0.1405	1.136	1.97	0.004153	1.061	0.0509	0.2428	+--
rs9938618	*MT3*	16	55184270	G	2.273	0.2868	1.296	3.987	0.004194	1.355	0.1269	0.01658	--+
rs2980213	*LYAR*	4	4336657	G	0.6374	0.1573	0.4683	0.8676	0.004203	0.990	0.0523	0.8504	-+-
rs325428	*BICD1*	12	32405069	C	1.491	0.1402	1.133	1.963	0.004386	1.005	0.0505	0.9188	+-+
rs7484762	*BICD1*	12	32394809	A	1.491	0.1402	1.133	1.963	0.004386	1.005	0.0505	0.9243	-+-
rs1678188	*SP100*	2	231000000	A	1.561	0.1573	1.147	2.124	0.004663	0.987	0.0621	0.8399	-++

**Table 4 genes-14-01029-t004:** Summary-level MR results with the replication meta-analysis data demonstrating the effect of telomere attrition on the risk of diabetic kidney disease (DKD) or end-stage kidney disease (ESKD) in T1D, using the Codd et al. IVs or the Park et al. IVs. The asterisks (*) identify that MR PRESSO reported significant pleiotropy (global test *p* = 0.0193); however, no variants presented significant horizontal pleiotropic effect in the outlier test and, therefore, the causal effect had no significant distortions. mF = mean F-statistic, unweighted I^2^, OR = odds ratio, CI = confidence interval.

IV Source	Genetically Predicted Exposure	Outcome	Number of Genetic Instruments Used	MR Egger Intercept *p*-Value	mF	unIsq	Summary-Level MR Method	OR	OR Lower CI	OR Upper CI	*p*-Value
Park et al./Li et al. [50]	Telomere attrition	DKD	21	0.63	46.97	0.886	Inverse variance weighted	1.04	0.61	1.78	0.883
							Simple median	0.81	0.32	2.03	0.649
							Weighted median	1.14	0.54	2.37	0.736
							MR Egger	1.38	0.39	4.94	0.622
							MR RAPS	0.98	0.53	1.84	0.959
							MR PRESSO	1.04	0.61	1.78	0.885
		ESKD	26	0.98	44.18	0.886	Inverse variance weighted	1.00	0.52	1.92	0.998
							Simple median	0.52	0.17	1.52	0.232
							Weighted median	2.23	0.90	5.50	0.082
							MR Egger	1.02	0.21	4.81	0.984
							MR RAPS	0.79	0.35	1.78	0.573
							MR PRESSO	1.00	0.52	1.92	0.998
Codd et al. [44]	Telomere length	DKD	53	0.08	164.83	0.983	Inverse variance weighted	0.90	0.60	1.37	0.630
							Simple median	0.63	0.29	1.34	0.227
							Weighted median	1.08	0.59	1.97	0.806
							MR Egger	1.51	0.75	3.07	0.256
							MR RAPS	0.96	0.63	1.46	0.851
							MR PRESSO	0.90	0.61	1.35	0.621
		ESKD	62	0.20	161.49	0.982	Inverse variance weighted	0.91	0.50	1.64	0.744
							Simple median	0.48	0.19	1.20	0.116
							Weighted median	1.19	0.53	2.66	0.671
							MR Egger	1.60	0.56	4.58	0.383
							MR RAPS	0.87	0.48	1.56	0.632
							MR PRESSO *	0.91	0.50	1.64	0.75

**Table 5 genes-14-01029-t005:** Top 15 genes presenting the highest number of CpG sites per genes significantly differentially methylated in diabetic kidney disease (DKD) compared to control individuals and the number of sites within these genes significantly associated with end-stage kidney disease (ESKD). Δβ = delta beta.

	DKD				ESKD			
Gene	Number of Significant CpG Sites	CpG Site Associated with Highest *p*-Value	Δβ	Highest Adjusted *p*-Value	Number of Significant CpG Sites	CpG Site Associated with Highest *p*-Value	Δβ	Highest Adjusted *p*-Value
*MAD1L1*	46	cg27238358	0.011	1.20 × 10^−13^	33	cg24247786	0.029	9.00 × 10^−14^
*TBCD*	16	cg26092192	−0.054	1.40 × 10^−14^	20	cg19882243	0.017	1.30 × 10^−11^
*BANP*	13	cg09975659	0.027	1.10 × 10^−16^	11	cg02604995	−0.056	4.90 × 10^−13^
*PFKP*	10	cg00593048	0.029	9.60 × 10^−13^	11	cg10348208	0.02	1.30 × 10^−13^
*TUBB*	10	cg02831473	0.02	5.50 × 10^−18^	8	cg00783730	−0.049	3.30 × 10^−12^
*WIPI2*	7	cg00987699	−0.084	7.40 × 10^−19^	8	cg00987699	−0.11	1.60 × 10^−17^
*AMPD2*	6	cg22164238	0.031	1.70 × 10^−17^	6	cg22164238	0.033	1.30 × 10^−12^
*HIST1H2AL*	6	cg05396178	0.11	4.00 × 10^−19^	5	cg19435409	0.047	2.80 × 10^−14^
*PA2G4*	6	cg14940636	0.087	6.50 × 10^−18^	3	cg14940636	0.088	7.90 × 10^−12^
*SRSF6*	6	cg12417590	−0.074	1.20 × 10^−18^	3	cg12417590	−0.087	1.60 × 10^−15^
*TERT*	5	cg10878976	0.01	1.70 × 10^−10^	6	cg17249224	0.013	1.80 × 10^−9^
*CTBP1*	5	cg20057475	0.02	9.80 × 10^−24^	6	cg14586363	−0.0079	1.10 × 10^−11^
*KLHDC4*	5	cg00782772	−0.022	8.70 × 10^−16^	5	cg00782772	−0.024	3.80 × 10^−12^
*TBCA*	5	cg23152667	−0.069	6.30 × 10^−20^	5	cg23152667	−0.061	5.60 × 10^−11^
*XPO1*	5	cg15711740	0.038	2.20 × 10^−15^	2	cg15711740	0.038	4.80 × 10^−10^

**Table 6 genes-14-01029-t006:** Genes from Table 5 which present associations with diabetes-related phenotypes, as determined via PheWEB.

Gene	Top *p* Value in Gene	Phenotype
*MAD1L1*	2.5 × 10^−7^	Type 2 diabetes
*TUBB*	1.3 × 10^−25^	Type 1 diabetes
	3.1 × 10^−12^	Diabetes mellitus
	1.5 × 10^−10^	Type 1 diabetes with ophthalmic manifestations
	1.6 × 10^−9^	Type 1 diabetes with ketoacidosis
*HIST1H2AL*	1.2 × 10^−15^	Treatment/medication code: insulin
*TBCA*	2.1 × 10^−9^	Diabetes mellitus in pregnancy

**Table 7 genes-14-01029-t007:** Genes which presented directional change in transcriptomic data which were concordant with the differential methylation in diabetic kidney disease (DKD) or end-stage kidney disease (ESKD) were associated with hypertension, diabetes, renal, glomerulonephritis or nephritis-related phenotypes in the UK Biobank.

Gene	Top *p*-Value in Gene	Phenotype
*AKR1B10*	2.3 × 10^−7^	Essential hypertension
	2.9 × 10^−7^	Hypertension
*ARRB1*	8.5 × 10^−7^	Type 2 diabetes with renal manifestations
*BAG3*	5.8 × 10^−7^	Cystic kidney disease
*CXCR4*	1.3 × 10^−6^	Renal failure NOS
*FANCA*	2.0 × 10^−8^	Essential hypertension
*HIST1H1A*	1.7 × 10^−10^	Hypertension
	2.8 × 10^−10^	Essential hypertension
*PACSIN1*	2.5 × 10^−10^	Hypertension
	3.6 × 10^−10^	Essential hypertension
*TUBB3*	2.0 × 10^−8^	Essential hypertension

## Data Availability

The GENIE cohort share genome-wide meta-analysis summary statistics (dbGaP Study Accession: phs000389.v1.p1). Additional data access requests can be made via https://www.ncbi.nlm.nih.gov/projects/gap/cgi-bin/study.cgi?study_id=phs000389.v1.p1 (accessed on 22 April 2023).

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
