# Peer review of "Differential Methylation of Telomere-Related Genes Is Associated with Kidney Disease in Individuals with Type 1 Diabetes"

_genes, 2023, doi:10.3390/genes14051029_

Round 1

Reviewer 1 Report

Hill et al. present a clearly written manuscript that provides a comprehensive analysis for the differeial methylation of telomere-related genes, combines genomic, epigenomic and trancriptomic data. This is an interesting research focusing on the connection among age, genetic variants, DNA methylation and gene expression, which is of immense value for the renal community. They found the telomere length difference related to age and also DKD. In particular, they identified hundreds of CpG sites associated with DKD and ESRD, which provide a useful resource to study the relationship between epigenetic changes, gene disfunction and kidney disease. This article should be of interest to many readers of Genes.

    Page 4 line 109. Please clarify the tissue (blood or kidney) which was used to extract the DNA for DNA methylation.
    Page 5 line 149. As telomere length is associated with both age and DKD, I am wondering what difference of telomere length between DKD and normal kidney with the same age range.
    Page 5 line 168. It would be better to look up the GWAS datasets mapped in the larger sample size.
    Page 10 line 265. “during diabetic kidney disease” should be “between DKD and normal”?
    Page 11 Figure 3. It would be perfect if you have any evidence for the association of CpG methylation and target gene expression.
    Page 13 Figure 6. Please include the statistic result for expression comparison between two groups for each gene.

Author Response

Hill et al. present a clearly written manuscript that provides a comprehensive analysis for the differential methylation of telomere-related genes, combines genomic, epigenomic and transcriptomic data. This is an interesting research focusing on the connection among age, genetic variants, DNA methylation and gene expression, which is of immense value for the renal community. They found the telomere length difference related to age and also DKD. In particular, they identified hundreds of CpG sites associated with DKD and ESRD, which provide a useful resource to study the relationship between epigenetic changes, gene disfunction and kidney disease. This article should be of interest to many readers of Genes.

We thank the Reviewer for the critical evaluation and positive feedback of our manuscript.

Page 4 line 109. Please clarify the tissue (blood or kidney) which was used to extract the DNA for DNA methylation.

We thank the Reviewer for highlighting this gap in information. We have now clarified in the text that the DNA methylation levels were derived from blood-derived DNA:

“Existing quantitative (blood-derived) DNA methylation levels were extracted for genes relevant to telomere function from the UK-ROI cohort”

Page 5 line 149. As telomere length is associated with both age and DKD, I am wondering what difference of telomere length between DKD and normal kidney with the same age range.

We thank the Reviewer for this excellent suggestion. P-values (Wilcoxon test and Bonferroni adjusted) have now been calculated between individuals with DKD and control individuals for each age group shown in Figure 2A. Significance was only observed in the 46 to 55 age group (P-adj = 0.0087). The following edits have been made in the text (section 3.1):

When stratifying by age group, a significant difference in telomere length in individuals with DKD, compared to control individuals, was only observed in the 46 to 55 age group (P=0.0087) (Figure 2A).

We have also added the number of individuals within each category to the legend of Figure 2:

The numbers of individuals in each age group are as follows, Under 26: N = 24, 26 to 35: N = 181, 36 to 45: N = 369, 46 to 55: N = 359 and 56 and above: N = 214.

Page 5 line 168. It would be better to look up the GWAS datasets mapped in the larger sample size.

We have now carried out additional look ups of the SNPs nominally associated with telomere length, harnessing the Common Metabolic Disease (CMD) portal (https://hugeamp.org/). We expanded the search region to 150kb either side of the SNP co-ordinate. This search showed that almost 70% of region investigated have previously been significantly associated with a range of renal and cardiovascular phenotypes. This CMD portal collates and analyses data from up to 398 datasets. We have added the following statement to section 3.2 to reflect these updated analyses, with the methods section 2.8. PheWAS analysis also updated accordingly:

Moreover, searching within 150kb of these SNP co-ordinates revealed that 25 of these regions (69.4%) were significantly associated with a range of renal or cardiovascular phenotypes in the Common Metabolic Disease portal database (clumped by linkage disequilibrium) (Table S5), which aggregates and analyses human genetic and functional genomic information linked to common metabolic diseases from up to 398 datasets.

We also utilised the same regions to explore the GWAS catalogue, to search for relevant phenotypes. A number of SNPs within these regions were associated with telomere length or DNA methylation, with the following addition now made to section 3.2 in the text:

Using these same regions to search the GWAS Catalogue, it was shown that the rs852540 SNP region (7:5,383,963-5,733,963) contained the variant rs7808152, previously associated with telomere length (P=1 x 10-6, Beta=- 0.1602507, CI = 0.096-0.225) in a cohort of 902 European ancestry individuals (Netherlands). It was also shown that within 150kb of variants rs2209437, rs2025557 and rs1536078 (all with the closest gene SH3GL2), variants associated with DNA methylation (rs7032102, P=2x10-8) and epigenetic clock age acceleration (GrimAge) in a cohort of up to 954 individuals from the UK (rs1114790, P=10-8, beta =-1.0232, CI=0.67-1.38) were identified.

The following addition was made the discussion (line 599) to highlight these findings:

The present study also highlighted a telomere-related gene (SH3GL2) which contained genetic variants nominally associated telomere length and genetic variants within 150 kilobases previously associated with DNA methylation or accelerated epigenetic ageing. Interestingly, SH3GL2 was significantly differentially methylated in ESKD in this study, and significantly dysregulated in both the Fan et al. and Levin et al. RNA-sequencing datasets, with decreased expression observed alongside (on average) decreased methylation.

Page 10 line 265. “during diabetic kidney disease” should be “between DKD and normal”?

We thank the Reviewer for their feedback and have altered the text to now read:

“in diabetic kidney disease (DKD) compared to control individuals”

Page 11 Figure 3. It would be perfect if you have any evidence for the association of CpG methylation and target gene expression.

We thank the reviewers for their feedback. We agree that evidence for the association between CpG methylation and target gene expression would provide value to this research field; however, this is an area that needs further work, as understanding of this association currently lacking. To address this question to the best of our ability, with resources available, we therefore harnessed external RNA sequencing data sets (by Levin et al. and Fan et al.) for DKD, which allowed genes of interest to be identified, where differentially methylation may affect gene expression in DKD.

Page 13 Figure 6. Please include the statistic result for expression comparison between two groups for each gene.

We thank the reviewer for highlighting this missing information. We have added adjusted p-values to Figure 6 A and B to represent the significance values obtained from the differential expression analysis. We have also streamlined Figure 6 C to make it easier to compare DKD and ESKD differential methylation profiles. Adjusted p-values from the differential expression analyses have also been added to Supplementary Figures 4, 5 and 6 where appropriate.

Reviewer 2 Report

In this study, Claire Hill and colleagues investigated genomic features affecting telomere length in patients with T1D and DKD in four independent White European populations. Authors reported that telomere length was associated with premature biological ageing in DKD. They found that telomere length was significantly shorter in older age DKD groups versus control individuals. The study demonstrated that DKD and ESKD were nominally associated with telomere-related genetic variation highlighting no significant association between genetically-predicted telomere length and kidney disease. More importantly, functional predictive analysis revealed differentially methylated genes that were closely involved in the Wnt signaling. Authors used numerous well established methods to carry out the study with sufficient population size (N=1147). Overall, the manuscript is well written and findings support the conclusions made by the authors.

Author Response

We thank this Reviewer for their positive feedback and their critical evaluation of this manuscript.

Reviewer 3 Report

This study explores the characteristics of accelerated aging as potential biomarkers or therapeutic targets for diabetic kidney disease (DKD). Using multi-omics data, the study examines features affecting telomere biology and any associated methylome dysregulation in DKD. The study finds that telomere length is significantly shorter in older age groups and significantly reduced in DKD versus control individuals, with significance remaining after covariate adjustment. DKD and end-stage kidney disease (ESKD) were also nominally associated with telomere-related genetic variation. The study identifies differentially methylated genes enriched for Wnt signaling involvement and potential targets where epigenetic dysregulation may result in altered gene expression, useful for potential diagnostic and therapeutic targets for intervention.

Overall, the conclusion is consistent with the findings of the study and provides a useful summary of the results. The study's multi-omics approach provides a comprehensive view of the underlying mechanisms of DKD and identifies potential targets for intervention. The study has some limitations, including a relatively small sample size, and further studies are needed to validate these findings. Despite these limitations, this study provides valuable insights into the potential role of accelerated aging as a biomarker or therapeutic target for DKD.

Dear authors, please:

1- some references cannot be found. (referencing manager Error!) Reference source not found.

2- is Figure 7. original? if no please take the copyrights 

Author Response

We thank this reviewer for their appraisal of our manuscript.

Dear authors, please:

1- some references cannot be found. (referencing manager Error!) Reference source not found.

Many thanks for highlighting this issue. This was a problem in Word to PDF conversion and it has now been rectified.

2- is Figure 7. original? if no please take the copyrights

Figure 7 is original and was developed using Biorender.com. Reference to this can be found in the Acknowledgments section: “Figure 7 and the graphical abstract were created using Biorender.com.”